# Mechanisms of Resistance to CDK4/6 Inhibitors and Predictive Biomarkers of Response in HR+/HER2-Metastatic Breast Cancer—A Review of the Literature

**DOI:** 10.3390/diagnostics13050987

**Published:** 2023-03-05

**Authors:** Ioana-Miruna Stanciu, Andreea Ioana Parosanu, Cristina Orlov-Slavu, Ion Cristian Iaciu, Ana Maria Popa, Cristina Mihaela Olaru, Cristina Florina Pirlog, Radu Constantin Vrabie, Cornelia Nitipir

**Affiliations:** 1Department of Oncology, “Carol Davila” University of Medicine and Pharmacy, 020021 Bucharest, Romania; 2Elias University Emergency Hospital, 011461 Bucharest, Romania

**Keywords:** CDK4/6 inhibitors, advanced/metastatic breast cancer, biomarkers of response, progression on CDK4/6 inhibitors, resistance mechanisms, endocrine therapy

## Abstract

The latest and newest discoveries for advanced and metastatic hormone receptor-positive (HR+) and human epidermal growth factor receptor 2-negative (HER2-) breast cancer are the three cyclin-dependent kinases 4 and 6 inhibitors (CDK4/6i) in association with endocrine therapy (ET). However, even if this treatment revolutionized the world and continued to be the first-line treatment choice for these patients, it also has its limitations, caused by de novo or acquired drug resistance which leads to inevitable progression after some time. Thus, an understanding of the overview of the targeted therapy which represents the gold therapy for this subtype of cancer is essential. The full potential of CDK4/6i is yet to be known, with many trials ongoing to expand their utility to other breast cancer subtypes, such as early breast cancer, and even to other cancers. Our research establishes the important idea that resistance to combined therapy (CDK4/6i + ET) can be due to resistance to endocrine therapy, to treatment with CDK4/6i, or to both. Individuals’ responses to treatment are based mostly on genetic features and molecular markers, as well as the tumor’s hallmarks; therefore, a future perspective is represented by personalized treatment based on the development of new biomarkers, and strategies to overcome drug resistance to combinations of ET and CDK4/6 inhibitors. The aim of our study was to centralize the mechanisms of resistance, and we believe that our work will have utility for everyone in the medical field who wants to deepen their knowledge about ET + CDK4/6 inhibitors resistance.

## 1. Introduction

Breast cancer has the highest number of new cases for both sexes and all ages, according to GLOBOCAN 2020. It is the second leading cause of mortality among women, and it has become a global health challenge. It is estimated that about 7.8 million women were diagnosed in 2021 [1]. Unfortunately, the global burden of breast cancer is increasing both in developed countries and in developing ones [2]. Breast cancer is grouped into four categories based on the immunohistochemical expression of hormone receptors: estrogen receptor positive (ER+), progesterone receptor positive (PR+), human epidermal growth factor receptor positive (HER2+), and triple-negative (TNBC), which is characterized by the lack of expression of any of the above receptors [3].

We found it of great interest and intriguing that one of the latest studies [4] on the regulation on signaling pathways, which highlighted that even natural products obtained from plants, fruits and vegetables (such as viridiflorol, verminoside, novel phloroglucinol derivatives, genistein, vulpinic acid, calcitrinone A, kaempferol, protopanaxadiol, thymoquinone, arctigenin, glycyrrhizin, 25-OCH3-PPD, oridonin, apigenin, wogonin, fisetin, curcumin, berberine, cimigenoside, and resveratrol) show anticancer activities against breast cancer through the inhibition of angiogenesis, cell migrations, proliferations, and tumor growth, as well as cell cycle arrest by inducing apoptosis and cell death, the downstream regulation of signaling pathways (such as Notch, NF-κB, PI3K/Akt/mTOR, MAPK/ERK, and NFAT-MDM2), and the regulation of EMT processes [4]. The investigators actually concluded that natural products also act synergistically to overcome the drug resistance issue, thus improving their efficacy as an emerging therapeutic option for breast cancer therapy. However, in this review we stay focused on molecular resistance to the treatment of HR+/HER2- breast cancer.

One of the most common subtypes (20–25% of all breast cancers) is HR+/HER2- breast cancer [5]. Endocrine therapy (ET) is the main treatment for the HR+ luminal subtype of breast cancer, in association with targeted therapy. Cyclin-dependent kinases 4 and 6 inhibitors (CDK4/6i) restore the cell cycle by selectively inhibiting cyclin-dependent kinases 4 and 6, and block cell proliferation in a variety of tumor cells, including those of breast cancer [6]. There are three CDK4/6 inhibitors approved by the US Food and Drug Administration that are transforming the treatment landscape nowadays: palbociclib, ribociclib, and abemaciclib (Table 1). They all have similar mechanisms of action and properties, with few differences in their preclinical and pharmacological settings and toxicity profiles [7]. There is a need for a personalized approach to overcome the growing financial burden for health care systems through more effective patient selection. Palbociclib, ribociclib and abemaciclib are expensive anticancer drugs because they are currently protected by drug patents, and hence the need for predictive biomarkers of response beyond estrogen receptor positivity [8].

## 2. Aims and Objectives

The management of breast cancer CDK4/6 inhibitor resistance is one of the most important clinical issues to be overcome, indicating a clear need for continuous discovery-based preclinical and clinical approaches. In order to assess these issues, we performed a systematic review of the published literature. The two key objectives were to identify resistance biomarkers and to understand molecular mechanisms underpinning drug resistance for CDK4/6 inhibition in breast cancer patients. Every single biomarker and signaling pathway was taken and discussed in separate paragraphs, highlighting the mechanism of possible resistance and its clinical and therapeutical implication.

## 3. Materials and Methods

The databases used to gather information for this review include Pubmed.gov and Clinicaltrials.gov. We reviewed the PubMed database from January 2013 to January 2023 and selected all relevant articles. The inclusion criteria for this literature review encompassed studies that examined resistance to CDK4/6 inhibitors. The inclusion criteria were studies that evaluated and validated biomarkers of predictive response to therapy and potential mechanisms of resistance. Studies that addressed future directions after the progression of inhibitors were also assessed. Exclusion criteria were articles with unavailable abstracts, non-English-written articles, and conference presentations. Keywords used to search for references included CDK4/6 inhibitor, biomarker, progression, and resistance in order to achieve the most specific results. The search generated 75 results, but only 25 articles met our criteria.

## 4. Review

### 4.1. How Do CDK4/6 Inhibitors Work?

The malignant transformation of normal cells begins with chaotic cellular proliferation, which takes place due to cell cycle dysregulation [9]. The cell cycle has four important stages: G1 (cells grow, increasing in size), S (synthesis of the DNA), G2 (cells grow more and make proteins), and M (mitosis). In the end, the cell splits into two daughter cells [10]. One of the most important cell cycle malfunctions starts right at the beginning of the cell cycle, which is controlled by the retinoblastoma protein (pRb). When in its active state, it stops the cell from progressing in the S phase by binding and suppressing E2F transcription factors. Phosphorylation of the Rb protein, which can be undertaken by the cyclin D–CDK4/6 complex, leads to E2F release. Thus, the cell can enter the S phase, and the cell cycle continues [11]. In turn, the complex is activated through the PI3K/AKT/mTOR and RAS/MAPK pathways by the activation of hormone receptors (including the estrogen receptor (ER)) and growth factors [12]. Obviously, the complex itself is downregulated by endogenous CDK inhibitors: the INK4 and Cip/Kip protein families [12]. A schematic representation of how CDK4/6 inhibitors work can be found in Figure 1.

There are several resistance mechanisms and potential biomarkers of response to CDK4/6 inhibitor regimens, which we will review in the upcoming paragraphs.

### 4.2. Cyclin D–CDK4/6 Abnormal Activation

The most frequently encountered resistance to CDK4/6i is the upregulation of the Cyclin D–CDK4/6–pRb pathway [13]. In a study conducted by Yang et al. in 2017, the majority of cells that were resistant to abemaciclib contained an amplification of CDK6 [14]. While CDK6 amplification was demonstrated to have an impact on potential treatment resistance, both high and low levels of CDK4 have been seen in resistance models [11].

In the same year, Gong et al. [15] demonstrated that cells with the highest sensitivity to abemaciclib showed increased cyclin D activity, which promotes cyclin D1 turnover [12]. Additionally, the overexpression of Cyclin D1 in breast cancer cells showed higher sensitivity to palbociclib [16]. However, many studies demonstrated that the overexpression of Cyclin D, with or without Cyclin D1 gene amplification, occurred in more than 50% of breast cancer cells [17]. Cyclin D1 is also a direct transcriptional target of ER [18], so the activation of the Cyclin D–CDK4/6 complex also contributes to endocrine therapy resistance [12].

Another important down-regulatory component of the complex is the p16 protein (a member of the CDKN2/INK family), whose inactivity could also contribute to aggressive breast cancer [17]. It is a tumor-suppressor protein that inhibits the activity of CDK4/6, and its expression correlates with a better prognosis in breast cancer patients. Low activity of p16 is correlated with increased CDK4/6 activity and increased sensitivity to palbociclib [19].

### 4.3. Loss of pRb

The loss of G1/S control is a hallmark of cancer, and is often caused by the inactivation of the retinoblastoma pathway [20]. As shown above, the integrity of the retinoblastoma protein is an important condition for the cells to be sensitive to CDK4/6 inhibitors, as it is at the center of the action mechanism. RB1 is the gene that encodes pRb, one of the most studied and reported biomarkers to date. Its loss or mutation is one of the most observed resistance mechanisms for CDK4/6i [21]. However, pRb function loss prior to CDK4/6i treatment is uncommon in metastatic breast cancer with HR+/HER2- [22]. In the PALOMA-3 study, only six out of 127 patients developed an RB1 loss of function after treatment with palbociclib and fulvestrant [23]. Another study conducted by Li et al. found a statistically significant difference in progression-free survival (PFS) regarding treatment with CDK4/6i; 3.6 months for patients who had a loss of the RB1 gene, compared to 10.1 months for patients with intact RB1 [16]. The first examples of acquired resistance were reported by Condorelli et al. [24], where acquired RB1 mutations were detected in ER-positive breast cancer patients treated with palbociclib and fulvestrant or ribociclib and letrozole.

To determine the function of Rb phosphorylation by cyclin D-CDK4/6, Topacio and colleagues sought to generate variants of Rb that could no longer interact with cyclin D-Cdk4,6 while preserving all the other interactions with other cyclin-Cdk complexes [25]. They analyzed the docking interactions between Rb and cyclin D-CDK4/6 complexes and found that cyclin D-CDK4/6 targets the Rb family of proteins for phosphorylation, primarily by docking a C-terminal alpha-helix, which is not recognized by the other major cell-cycle cyclin-CDK complexes cyclin E-CDK2, cyclin A-CDK2, and cyclin B-CDK1 [25]. Their results showed that cyclin D-CDK4/6 phosphorylates and inhibits Rb via a C-terminal helix, and that this interaction is a major driver of cell proliferation [25].

### 4.4. Cyclin E–CDK2 Pathway Activation

During a normal cell cycle, cyclin E1 and cyclin E2 can bind to and activate CDK2 in order to phosphorylate pRb, but only after it has already been phosphorylated by the cyclin D–CDK4/6 complex as a second wave of signaling [11]. The activation of the cyclin E1/cyclin E2-CDK2 complex permits cells to bypass the inhibiting activity of CDK4/6 and encourage growth and proliferation [13]. Therefore, the overexpression of cyclin E1, cyclin E2, and CDK2 can subvert the CDK4/6 inhibition [11].

An interesting study conducted by Guarducci et al. showed that the ratio of cyclin E1 to RB1 level (not only cyclin E1 amplification and RB1 loss) is a poor prognostic factor and predicts palbociclib de novo resistance in HR+ breast cancer [26]. Herrera-Abreu et al. demonstrate in a study from 2016 that cyclin E1 is upregulated via CDK2 activation in palbociclib-resistant cells (that were generated via chronic exposure to the drug and named palbociclib-resistant MCF-7 breast cancer cells) [27]. In a phase II study (the NeoPalAna trial), researchers studied palbociclib resistance in patients with high levels of cyclin E1 [28]. Cyclin E1 overexpression was also predictive of an abemaciclib response to targeted therapy, as shown in the study conducted by Gong X et al. [29].

Next, gene expression analysis of 302 ER+ breast cancer samples from PALOMA-3 trial revealed that lower Cyclin E1 (CCNE1) mRNA levels were associated with a better response to palbociclib [30]. This association was confirmed in a preoperative setting, in the cohort of POP (PreOperative Palbociclib) trial [31].

Taking all this together, cyclin E1, cyclin E2, and CDK2 are upregulated in the CDK4/6 inhibitor resistance models [11].

### 4.5. PI3K/AKT/mTOR Pathway Activation

This signaling pathway activation is another mechanism for both de novo and acquired resistance to CDK4/6i, with the hyperactivity of PI3K playing a role in endocrine-resistant mechanisms [17]. PIK3CA mutations could be identified in almost 40% of breast cancers with hormonal receptors [32]. Activating PIK3CA mutations could be a biomarker of either intrinsic resistance or acquired resistance. However, PI3KCA mutations have not been associated with resistance to CDK inhibitors in clinical studies to date [12].

One study identified that the PI3K pathway kinase (PDK1) was overexpressed in ribociclib-resistant cells [21]. Not only in ribociclib-resistant cell lines, but also in palbociclib-resistant cell lines, PIK3CA loss led to reduced proliferation of all cell lines regardless of RB status, as shown by Attia and colleagues in a 2020 study [21,33].

There are works in the literature that suggest adding a PI3K inhibitor, such as alpelisib, to CDK4/6i in order to circumvent the resistance mechanisms that develop for CDK4/6. It could be added after progression on CDK4/6i and ET (endocrine therapy), or from the start in triple combination to prevent the onset of resistance to the combination of CDK4/6i and ET (via modulation of early adaptive response) [34].

The mammalian target of rapamycin (mTOR) is implicated in cell cycle processes such as cell growth, size control, division, and proliferation, and it could be one of the reasons for CDK4/6i resistance. mTORC1 and mTORC2 are two different complexes that are formed by the mTOR kinase. A study conducted by Michaloglou and colleagues demonstrates that an mTORC1/mTORC2 inhibitor (vistusertib) could prevent early adaptive resistance to palbociclib in HR-positive breast cancer cells [35]. According to the specialty literature, the most frequent therapy used after progression on CDK4/6i is the mTOR inhibitor (everolimus) [36].

The AKT (serine/threonine kinase of the AGC kinase family) is activated via phosphorylation, which induces growth and survival. For this process, PDK1 (3-phosphoinositide dependent kinase 1) has an important role in the PI3K–AKT pathway. A low level of PDK1 makes tumor cells more sensitive to CDK4/6i [37]. On the other hand, a high level of AKT1 activity was seen in palbociclib-resistant cells [38].

### 4.6. FGFR1 Activation (FGF/FGFR Signaling Pathway Activation)

Fibroblast growth factor receptor 1 (FGFR1) is a protein of the tyrosine kinase family that plays an important role in the cell cycle, being implicated in the migration, proliferation, differentiation, and survival of the cells. In more than 15% of breast cancers with hormone receptors present, a mutation of FGFR1 is found [39]. Thus, the causal relationship between FGFR1 mutations and endocrine therapy resistance has already been explained and demonstrated [13]. It is also important to find out if there is a connection between these mutations and resistance to CDK4/6 inhibitors. In order to do this, Formisano and colleagues showed that the cells that overexpressed FGFR1 were resistant to ribociclib and fulvestrant, and they also demonstrated that the cells that received an FGFR1 tyrosine kinase inhibitor (lucitanib) reversed the resistance. Moreover, the study highlighted a shorter PFS rate in those with FGFR overexpression among patients enrolled in the MONALEESA-2 clinical trial [38]. Surprisingly, the patients enrolled in the PALOMA-2 trial with FGFR2 amplification in the palbociclib + letrozol arm benefited from a longer PFS than those who were given placebo and letrozole [40].

In a study from 2019, Drago and colleagues [39] evaluated the clinical response to endocrine and targeted therapies in a cohort of 110 patients with HR+/HER2− metastatic breast cancer and validated the functional role of FGFR1-amplification in mediating response/resistance to hormone therapy in vitro. The investigators concluded that, while FGFR1 amplification confers broad resistance to ER, PI3K, and CDK4/6 inhibitors, mTOR inhibitors might have a unique therapeutic role in the treatment of patients with ER+/FGFR1+ metastatic breast cancer [39]. Another study conducted by Mouron et al. [41] included 251 patients with HR+ breast cancer and studied the role of ER, CDK4/6, and/or FGFR1 blockade alone or in combinations in Rb phosphorylation, cell cycle, and survival. They showed how hormonal deprivation leads to FGFR1 overexpression, thus being associated with resistance to hormonal monotherapy or in combination with palbociclib. Both resistances have been reverted with triple ER, CDK4/6, and FGFR1 blockade [41].

### 4.7. RAS Activation

The RAS family of protooncogenes encodes three oncogenes, KRAS, NRAS, and HRAS, each with important roles in the cell cycle, such as apoptosis, growth, and differentiation. Kirsten Rat Sarcoma Viral Oncogene Homolog (KRAS) is the most frequently mutated RAS gene [13]. Many studies over the last years have revealed how the engagement of RAS function might result in mandatory downstream varied oncogenic alterations for progression, metastatic dissemination, and therapy resistance in breast cancers [42]. In this direction, we found a review from 2019 conducted by Galie where he underlined the major studies over the last 30 years which have explored the role of RAS proteins and their mutation in breast cancer patients [42].

An overexpression of KRAS has been associated over the years with many types of cancer growth and development, including breast cancer resistance to CDK4/6i. A study from 2021 by Raimondi et al., who enrolled 106 patients with HR+ metastatic breast cancer, showed resistance to palbociclib and fulvestrant in the cells that developed a KRAS amplification. Moreover, the PFS was just three months for the subjects with KRAS mutations, whereas, in the other arm, the PFS had not even been reached by the 18-month follow-up [8,13]. Cells with KRAS, NRAS, and HRAS activating mutations are, therefore, susceptible to CDK4/6 inhibitor resistance [8,13].

### 4.8. FAT1 Loss

Another important and well-studied biomarker of possible resistance to CDK4/6i is the loss of FAT1. FAT atypical cadherin 1 (FAT1) is among the most frequently mutated genes in many types of cancer [43]. This is a tumor suppressor gene, a member of the cadherin superfamily, which interacts with beta-catenin and Hippo signaling pathways. It is found in 6% of metastatic HR+ breast cancers [44]. Chen and colleagues performed a literature review on the diverse functions of FAT1 in cancer progression and presented the phenotypic alterations due to FAT1 mutations, several signaling pathways and tumor immune systems known or proposed to be regulated by this protein [43]. A study conducted by Li et al. on 348 patients treated with CDK4/6 inhibitors highlighted, after genetic sequencing, that patients with loss of FAT1 had a lower PFS compared to those with intact FAT1 (2.4 months and 10.1 months, respectively) and rendered cells resistant to all three CDK4/6i. The investigators highlighted that FAT1 loss is also associated with CDK6 overexpression via downregulation of the Hippo signaling pathway (through YAP and TAZ transcription factors) [45]. The role of FAT1 deleterious mutations was then confirmed in vivo. Cells with FAT1 knockout or knockdown did not stop cell growth upon exposure to abemaciclib, and MCF7-implanted xenografts experienced much less sensitivity to abemaciclib than mice with a non-mutated FAT1 gene [13,21].

### 4.9. PTEN Loss

PTEN is a tumor suppressor gene and one of the frequently mutated genes in human cancers [46]. The increased expression of PTEN leads to the inactivation of CDK, which enables the Rb1 to keep dephosphorylating, while binding to transcription factor E2F, which ultimately inhibits cell proliferation [47]. The overexpression of AKT could reduce PTEN expression and render breast cancer cells resistant to CDK4/6i [48]. Costa and colleagues performed an analysis of serial biopsies, which uncovered both RB and PTEN loss as mechanisms of acquired resistance to CDK4/6i. The investigators demonstrated that, in breast cancer cells, the ablation of PTEN through increased AKT activation was sufficient to promote resistance to CDK4/6 inhibition (ribociclib and letrozole) in vitro and in vivo; PTEN loss resulted in the exclusion of p27 from the nucleus, leading to increased activation of both CDK4 and CDK2 [49]. PTEN loss is rare in treatment-naïve ER-positive tumors [50,51]. The loss of PTEN confers resistance to PI3K inhibitors (alpelisib) [48], as well as cross-resistance to CDK4/6i and PI3K inhibitors [52]. Lee and colleagues conducted a retrospective analysis using real-world data, molecular biomarkers such as FGFR1 amplification, PTEN loss, and DNA repair pathway gene mutations, and showed a significant association of shorter PFS with CDK4/6i therapy [53].

### 4.10. S6K1 Amplification

S6K1 is a conserved serine/threonine protein kinase that belongs to the family of protein kinases, being the principal kinase effector downstream of the mammalian target of rapamycin complex 1 (mTORC1) [54]. S6K1 is an important regulator of cell size control, protein translation and cell proliferation [55]. S6K1 is one of the best-characterized downstream targets of mTORC1, and rapamycin treatment results in rapid dephosphorylation and inactivation of S6K1 [56]. The hyperactivation of mTORC1/S6K1 signaling may be closely related to ER-positive status in breast cancer, and may be utilized as a marker for prognosis and a therapeutic target [54]. A study from 2012 highlights that the S6K1–ER relationship creates a positive feed-forward loop in the control of breast cancer cell proliferation and, furthermore, the co-dependent association between S6K1 and ERα may be exploited in the development of targeted breast cancer therapies [57]. During the literature review, we found of interest a recent research article from August 2022 conducted by Mo and colleagues [58] regarding S6K1 amplification. The Chinese investigators demonstrated that S6K1 amplification confers innate resistance to palbociclib and ET through activating c-Myc pathway in 36 patients with ER+ breast cancer. In those who had received palbociclib, patients with high-expressed S6K1 had significantly worse progression-free survival and significantly worse relapse-free survival than those with low S6K1 expression. S6K1 overexpression was sufficient to promote resistance to palbociclib. S6K1 overexpression increased the expression levels of G1/S transition-related proteins and the phosphorylation of Rb, mainly through the activation of the c-Myc pathway. Mo et al. showed that this resistance could be abrogated by the addition of the mTOR inhibitor, which blocked the upstream of S6K1, in vitro and in vivo [58].

### 4.11. AURKA Amplification

Aurora kinase A (AURKA) belongs to the family of serine/threonine kinases, whose activation is necessary for cell division processes via the regulation of mitosis. AURKA shows significantly higher expression in cancer tissues than in normal control tissues for multiple tumor types [59]. The amplification of the mitotic kinase AURKA has been identified in 11 out of 41 HR+ breast cancer biopsies from tumors resistant to CDK4/6 inhibitors, including examples of both intrinsic and acquired resistance, with no alterations detected in sensitive samples [60]. Aurora A has been previously shown to mediate endocrine resistance through the downregulation of ER expression in an SMAD5-dependent manner [61]. Two studies have shown that Aurora kinase A/B inhibition is synthetically lethal with RB1 deficiency in breast cancer and small-cell lung cancer cell lines [62,63], suggesting alternative therapeutic strategies for RB1-null tumors or new combinatorial strategies to prevent acquired resistances to CDK4/6 inhibitors [57].

### 4.12. c-Myc Upregulation

c-Myc is a member of a family of protooncogenes that code for transcription factors, and is often overexpressed in cancer [64]. It is activated by phosphorylation, and in this form c-Myc is stable and allows cells to escape senescence. CDK2 and CDK4/6 inhibition decreases the phosphorylation of c-Myc, which destabilizes the gene and allows cells to enter the apoptosis process [13]. Mateyak et al. performed a comprehensive analysis and found that the largest defect observed in c-myc-/- cells was a 12-fold reduction in the activity of cyclin D1-CDK4/6 complexes during the G0 to S transition. The investigators suggested that c-Myc affects the cell cycle at multiple independent points, because the restoration of the CDK4 and 6 defect does not significantly increase growth rate [65].

Pandey et al. concluded in a study from 2020 that overexpression of c-Myc leads to palbociclib-resistant cells [66]. In the MONARCH-3 trial, 5% of patients with newly acquired c-Myc mutations were associated with resistance to abemaciclib + ET, and 9% of patients treated with abemaciclib alone in the MONARCH-1 trial acquired new Myc alterations [13,67].

### 4.13. miR Downregulation

MicroRNAs are non-coding RNA molecules involved in the post-transcriptional regulation of gene expression and regulate 30–60% of the human genome. MicroRNAs regulate the cell cycle through cyclin-dependent kinases and cyclins. The downregulation of miRNAs negatively regulates CDK6, which leads to CDK6 activation. CDK6 activation results in palbociclib resistance, as shown by Li and colleagues in a study from 2020 [44,68]. Moreover, in a retrospective analysis of 44 patients treated with CDK4/6i, microRNA levels were higher in those with intrinsic or acquired CDK4/6i resistance [69].

Krasniqi et al. summarized in their study that some miRNAs (such as miR-326, miR-29b-3p, miR-126, and miR3613-3p) are associated with sensitivity to CDK4/6 inhibitors, whereas others (such as miR-432-5p, miR-223, and miR-106b) appear to confer treatment resistance [70]. Identifying specific expression patterns of miRNAs could be a promising approach to study tumor response to CDK 4/6 inhibitors and exploit them as novel biomarkers [70]. Non-coding RNAs have been demonstrated to be strictly lineage-specific; their expression may therefore determine cell phenotype, allowing for the identification of specific tumor sub-populations resistant to CDK inhibitors [71].

### 4.14. TK1 Activity

Thymidine kinase 1 (TK1) is a DNA salvage pathway enzyme involved in regenerating thymidine for DNA synthesis and DNA damage [72]. It catalyzes the conversion of thymidine to deoxythymidine monophosphate, which is further phosphorylated to di- and triphosphates before its use for DNA synthesis [73]. In resting cells, observable TK1 activity is low to absent, increasing during G1/S transcription and peaking at S phase [74]. In healthy subjects, levels of TK1 are low to absent, with contrastingly elevated levels observed in patients with a range of malignancies, including breast cancer [75]. TK1 is a phosphotransferase that plays a role in DNA replication, is regulated by the E2F pathway, and is downstream of CDK4/6. Its activity is a marker of tumor proliferation. TKs’ activity has been shown to be a prognostic marker in patients with metastatic breast cancer, both when measured at baseline and during treatment. There are some clinical studies that support this statement [36,44].

A prospective monitoring trial (ClinicalTrials.gov NCT01322893) from Sweden, in which 156 metastatic breast cancer patients planned to start first-line systemic therapy, has reported that the TK1 activity level is prognostic for survival (decreases in TK1 levels from 3 to 6 months correlate to improved survival PFS and OS) in patients with newly diagnosed metastatic breast cancer [76].

McCartney and colleagues reported that intense TK1 activity is seen in cell lines resistant to palbociclib. The phase II TRend study also reported a shorter PFS for patients with high levels of TK1 than in the other arm [22,66] (3 months vs. 9 months) [77]. Another study (the ECLIPS trial) reported progressive disease in patients with metastatic breast cancer treated with palbociclib [78]. In the NeoPalAna trial, investigators observed an important reduction in TK1 activity after the initiation of palbociclib, suggesting a reduction in tumor proliferation [28,79].

### 4.15. Endocrine Resistance and CDK4/6i Sensitivity—An Association Worthy of Consideration

Endocrine treatment is one of the most important approaches when it comes to ER+ breast cancers, and for metastatic disease it becomes the physician’s first choice, along with other targeted therapies (except in the case of a visceral crisis scenario, when chemotherapy should be the first choice). To date, some endocrine-resistant mechanisms have been described, including the upregulation of ER coactivators (e.g., FOXA1), cyclins (cyclin D and E), CDK proteins (CDK2 and CDK6), mitogen signaling pathways (PI3K and RAS pathways), or the downregulation of CDK inhibitor proteins (p16) [11]. As already known, CDK4/6 inhibition acts downstream of endocrine therapy; therefore, some resistance mechanisms are common to both types of treatments (endocrine therapy and CDK4/6 inhibitors) [11].

Among these resistance mechanisms, many studies and clinical trials have found a connection between estrogen receptor 1 (ESR1) mutations and acquired resistance to endocrine therapy. ESR1 mutations are the most important alterations resulting in resistance to aromatase inhibitor treatment, and can be found in almost 40% of metastatic breast cancer patients [80] and in approximately 20% of patients with endocrine-resistant breast cancer [81]. However, no association was found between ESR1 and CDK4/6i resistance. In the MONALEESA-2 trial, there was no correlation between ESR1 levels and response to ribociclib [80], and neither was there in the PALOMA-3 trial, where there was no link between ESR1 mutations and response to palbociclib [9]. Moreover, the PFS was improved both for patients with ESR1 mutations and for patients with non-mutated ESR1, demonstrating that this mutation does not affect treatment response. However, in the PALOMA-3 trial, at the end of the treatment 12.8% of patients developed new mutations in the ESR1 gene, with the Y537S mutation in particular [12]. Different results were observed in MONARCH-2, in which patients with ERS1 mutations showed an overall survival benefit [82]. O’Leary and colleagues also investigated PIK3CA mutations and concluded that both PIK3CA and ESR1 mutations were evenly distributed in both arms of the study, which leads to the idea that these mutations are more likely to affect the response to fulvestrant than to palbociclib [23]. The PALOMA-3 trial highlights the idea that ET resistance should be taken into consideration when talking about resistance to combination regimens in HR+/HER2-breast cancer. There is also an ongoing trial from Johns Hopkins University (NCT03439735) that studies the association between ESR1 mutations and clinical outcomes in patients treated with palbociclib and aromatase inhibitor as a first-line treatment regimen; its reported results should be available in June 2024.

However, all three pivotal clinical trials (PALOMA-3, MONARCH-2, and MONALEESA-3) demonstrated that CDK4/6 inhibitors prolong PFS even after ET resistance, which demonstrates that CDK4/6i maintain effectiveness regardless of the endocrine-resistant disease. Additionally, endocrine-resistant tumors maintain sensitivity to CDK4/6 inhibitors, particularly when they are used in association with ET [11].

## 5. Discussions

CDK4/6 inhibitors remain a landmark for the treatment of hormone receptor-positive and human epidermal growth factor receptor 2-negative metastatic breast cancer, being the most significant advance in the last decade. Various preclinical and translational research efforts have begun to shed light on the genomic and molecular landscape of resistance to these agents [83]. As we showed above, it is important to understand the mechanism of action of CDK4/6 inhibitors in order to target specific signaling pathways and predictive biomarkers of response, taking into consideration that intrinsic and acquired resistance could limit the activity of these inhibitors. In addition, one of the greatest challenges is distinguishing between mechanisms causing resistance to CDK4/6 inhibition and endocrine resistance.

Approximately 10% of patients will have primary resistance to CDK4/6 inhibitors [84]. For instance, patients with evidence of functional Rb loss at baseline are not likely to benefit from CDK4/6 inhibition, or from increased cyclin E1/E2 expression. A rise in TK1 activity may also provide a marker of early resistance [84]. Mutations in RB1, resulting in the activation of other cell cycle factors, such as E2F and the Cyclin E-CDK2 axis, have been demonstrated in cases of acquired resistance [84]. In the table below (Table 2), we summarized the main resistance mechanisms and biomarkers of resistance, which we have previously reviewed.

Following progression, no prospective randomized data exist to help guide second-line treatment [85]. While prospective data are needed, analysis of real-world data suggests a survival benefit for the continuation of CDK4/6i beyond a frontline progression for patients with HR+/HER2- metastatic breast cancer [85]. Several ongoing Phase 1 and 2 trials (MAINTAIN NCT02632045, PACE NCT03147287, NCT01857193, NCT 02871791, and TRINITI-1 NCT 02732119) are investigating the potential benefit of continuing CDK4/6i beyond progression [84]. For more successful treatment, biomarkers are of potential interest in order to identify patients who might be responsive or not to CDK4/6 inhibitors, facilitating an early switch to a more efficacious treatment.

## 6. Conclusions

To date, no biomarker has been studied enough to be approved as a predictor of response to treatment or a targeted signaling pathway. Personalized treatment based on an individual’s response and tumor genomics represents the future of oncology. Therefore, it is a justification for future clinical trials because the identification of biomarkers of resistance is still a problem universally, and there is still more to be discovered about CDK4/6 inhibitor resistance. The optimum management of HR+/HER2-metastatic breast cancer is essential for patients as they might have only one more card to play, so future therapeutic targets should be examined in clinical trials to delay or overcome treatment resistance to combinations of ET and CDK4/6 inhibitors.

In conclusion, we strongly believe that the validation of proposed biomarkers should be an option to consider before starting treatment with CDK4/6 inhibitors and hormonal therapy. This can be carried out via whole exome and targeted sequencing of solid and liquid biopsies, in order to reveal several possible genomic alterations that could change the course of treatment. In Romania, unfortunately there are few patients who can afford the costs of this type of testing. After doing such exhaustive research for this review, our personal opinion is that some biomarkers are worth testing more than others, such as loss of retinoblastoma protein. Some mechanisms of resistance, such as PI3K/AKT/mTOR or Cyclin E–CDK2 pathway activation, have already had their implication validated in resistance to CDK4/6i + ET; therefore it would be a worthy idea to take into consideration before starting the treatment. Breast cancer patients, maybe more than any other patients, are susceptible to depression and self-esteem loss, thus making any kind of treatment more difficult. We believe that a good start is always a better start and we do hope that in the near future breast cancer patients would benefit from the best personalized treatment.

## Figures and Tables

**Figure 1 diagnostics-13-00987-f001:**
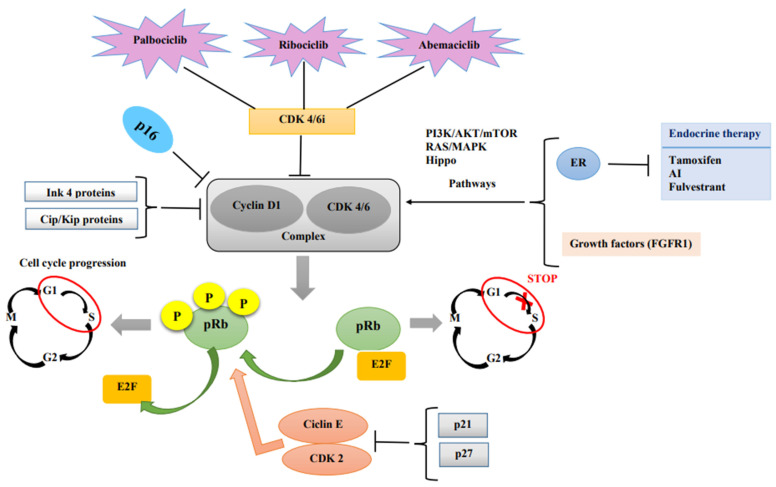
Key mechanisms of action of CDK4/6 inhibitors in HR+/HER2- breast cancer.

**Table 1 diagnostics-13-00987-t001:** Pivotal studies on the three approved CDK4/6 inhibitors.

	Study	Description	Phase	Number of Patients Enrolled	Median PFS	Identifier NCT	Status	Treatment
1st line	PALOMA-1	Palbociclib + Letrozol vs. Placebo + Letrozole	II	177	18.1 m vs. 11.1 m	NCT00721409	Completed	Palbociclib 125 mg/d orally for 3 weeks, followed by 1 week off Letrozole 2.5 mg/d orally on a continuous regimen Placebo 125 mg/d orally for 3 weeks, followed by 1 week off
	PALOMA-2	Palbociclib + Letrozol vs. Placebo + Letrozole	III	666	24.8 m vs. 14.5 m	NCT01740427	Active, not recruiting	Palbociclib 125 mg/d orally for 3 weeks, followed by 1 week off Letrozole 2.5 mg/d orally on a continuous regimen Placebo 125 mg/d orally for 3 weeks, followed by 1 week off
	MONALEESA-2	Ribociclib + Letrozol vs. Placebo + Letrozole	III	668	25.3 m vs. 16.0 m	NCT01958021	Active, not recruiting	Ribociclib 200 mg × 3/d orally for 3 weeks, followed by 1 week off Letrozole 2.5 mg/d orally on a continuous regimen Placebo 200 mg × 3/d orally for 3 weeks, followed by 1 week off
	MONALEESA-7	Ribociclib + Goserelin + Tamoxifen/Letrozole/Anastrozole vs. Placebo +Goserelin +Tamoxifen/Letrozole/Anastrozole	III	672	23.8 m vs. 13.0 m	NCT02278120	Active, not recruiting	Ribociclib 200 mg × 3/d orally for 3 weeks, followed by 1 week off Goserelin 3.6 mg subcutaneous injection once every 28 days Letrozole 2.5 mg/d orally or Anastrozole 1 mg/d orally or Tamoxifen 20 mg/d orally on a continuous regimen Placebo 200 mg × 3/d orally for 3 weeks, followed by 1 week off
	MONARCH-3	Abemaciclib + Letrozole/Anastrozole vs. Placebo + Letrozole/Anastrozole	III	493	28.18 m vs. 14.76 m	NCT02246621	Active, not recruiting	Abemaciclib 150 mg × 2/d orally Letrozole 2.5 mg/d orally or Anastrozole 1 mg/d orally Placebo 150 mg × 2/d orally
2nd line	PALOMA-3	Palbociclib + Fulvestrant vs. Placebo + Fulvestrant	III	521	9.50 m vs. 4.60 m	NCT01942135	Completed	Palbociclib 125 mg/d orally for 3 weeks, followed by 1 week off Fulvestrant 500 mg intramuscular injection on day 1 and day 15 of cycle 1 and then on day 1 of each cycle Placebo 125 mg/d orally for 3 weeks, followed by 1 week off
	MONARCH-2	Abemaciclib + Fulvestrant vs. Placebo + Fulvestrant	III	669	16.40 m vs. 9.30 m	NCT02107703	Active, not recruiting	Abemaciclib 150 mg × 2/d orally Fulvestrant 500 mg intramuscular injection on day 1 and day 15 of cycle 1 and then on day 1 of each cycle Placebo 150 mg × 2/d orally
Later line	MONARCH-1	Abemaciclib alone (one arm clinical trial)	II	132	5.95 m	NCT02102490	Completed	Abemaciclib 200 mg × 2/d orally

**Table 2 diagnostics-13-00987-t002:** The main resistance mechanisms and biomarkers of resistance to CDK 4/6 inhibitors.

Activation	Downregulation	ET Resistance
Cyclin E1, E2	RB	ESR1 mutations
PI3K/AKT/mTOR	FAT1	
AKT1	PTEN	
CDK2	miR	
CDK6		
FGFR1		
MYC		
RAS		
AURKA		
S6K1		

## Data Availability

Not applicable.

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
