# Peer review of "Mechanisms of Resistance to CDK4/6 Inhibitors and Predictive Biomarkers of Response in HR+/HER2-Metastatic Breast Cancer—A Review of the Literature"

_diagnostics, 2023, doi:10.3390/diagnostics13050987_

Round 1

Reviewer 1 Report

1. Figure 2 should be improved the clarity 

2.  Table 2 should include before the conclusion. it may be a paragraph. 

3. 

Reviewer 2 Report

Comments to Authors

1-Please mention the future perspectives in the abstract section

2-Please ad the appropriate keywords with 5-6

3-Line 20: Replace the sentence with appropriate sentence.

4-Introduction is too short. Please ad 2-3 more paragaraghs in introduction section. Authors can take some information from latest article doi.org/10.3390/molecules27113412

5- Please ad the aims and objectives as one paragraph. Authors should describe the aims of the study in one paragraph.

6-Citations are missing sources in Line 147. Please mention the citation source.

7-Please write one more paragraph about TK1 activity as it reflected the major part of mechanism.

8-Please headings the all numberings throughout the manuscript.

9- It is recommended to update Conclusion it reflected the major study concerns.

Reviewer 3 Report

In this mini-review, the author discussed mechanisms of resistance to CDK4/6 Inhibitors and predictive biomarkers of response in HR+/HER2-metastatic breast cancer. This summarized literature from January 2013 to January 2023 and concluded the main resistance mechanisms and biomarkers of resistance to CDK 4/6 inhibitors. By discussing the various pathways as follows

Cyclin D–CDK4/6 abnormal activation; Loss of pRb; Cyclin E–CDK2 pathway activation; PI3K/AKT/mTOR pathway activation; FGFR1 activation (FGF/FGFR signaling pathway activation); RAS activation; FAT1 loss; c-Myc upregulation; miR downregulation; TK1 activity; Endocrine resistance and CDK4/6i sensitivity – an association worthy of consideration.

While investigating the literature found an article in Front. Oncol. (https://doi.org/10.3389/fonc.2019.00666) the similar review on Mechanisms of Resistance to CDK4/6 Inhibitors and this is not cited in the current manuscript. This reviewer recommends to the authors verify the above reference and then update the current review article with highlights.

The current manuscript looking need to update with more literature survey and some sections need improvement in the explanation of the mechanisms.

Round 2

Reviewer 2 Report

No

Reviewer 3 Report

Responses from the authors were convincing and comparatively improved the current form of the article. The current manuscript with major revisions; the Introduction updated with the background of the recent cause of breast cancer and additionally added three different mechanistic pathways; PTEN loss, S6K1 amplification, and AURKA amplification. The literature was also updated accordingly. Overall this reviewer is satisfied and recommended for publication.